# DIFFERENTIABLE AVERAGE PRECISION LOSS IN DETR

## ABSTRACT

Average Precision (AP) is a widely used metric for evaluating object detection systems because it effectively integrates both classification accuracy and localization precision. In this paper, we conduct a detailed analysis of the characteristics of the AP metric, focusing on its non-differentiability and non-convexity. Building on this analysis, we propose a novel loss function called Differentiable Average Precision Loss (DAP-loss), which provides a differentiable approximation of AP, thereby enabling direct optimization of AP across a set of images. We validate the effectiveness of DAP-loss both theoretically and empirically, extending its application to the cost functions used in the Hungarian matching algorithm, which makes it suitable for end-to-end detection models. DAP-loss supports the simultaneous optimization of classification and localization tasks within an end-to-end framework, eliminating the need for hyperparameters to balance these tasks—a common challenge in previous methods. In the later stages of training, we applied DAP-loss to replace the original loss functions in several state-of-the-art end-to-end models, including DETR (Carion et al., 2020) and Deformable DETR (Zhu et al., 2020). Experimental results demonstrate that our method achieves significant improvements over baselines on the COCO dataset.

## 1 INTRODUCTION

Object detection is a crucial task in computer vision, focused on identifying and localizing objects within images. It requires optimizing both classification accuracy and the precise localization of object boundaries. Most existing deep learning-based object detectors (*e.g.* Carion et al. (2020); Ren et al. (2015); Redmon et al. (2016)) rely on multi-task learning frameworks, using separate loss functions to address classification and localization issues independently. Although this method enables straightforward model training via gradient descent, it introduces multiple hyperparameters to balance the weights of different losses, resulting in significant costs associated with hyperparameter tuning. In contrast, our proposed DAP-loss involves only a single hyperparameter that requires tuning, making it much easier to fine-tune for optimal performance.

Average Precision (AP) is a pivotal evaluation metric in object detection, integrating both classification accuracy and localization precision. Despite its widespread use, the non-differentiable and non-convex nature of AP poses significant challenges for direct optimization using gradient descent techniques. Various methods, such as Henderson & Ferrari (2017a); Chen et al. (2019); Mohapatra et al. (2018), have been proposed to try to directly optimize AP in the field of object detection. However, most of these approaches primarily focus on score ranking or classification tasks, often neglecting the localization aspect. As a result, these methods require additional loss functions and hyperparameters to achieve comprehensive optimization. Thus, the optimization of AP-loss remains an open problem.

In this paper, we provide a comprehensive analysis of the AP metric, examining both its inherent characteristics and the difficulties associated with its gradient, which may be non-existent or zero. Building on this analysis, we introduce a novel differentiable approximation of AP, termed Differentiable Average Precision Loss (DAP-loss), aimed at addressing these limitations. Specifically, we decompose AP into the product of a localization function and a classification function. For localization, we use interpolation techniques to create a smooth and differentiable representation

of localization errors. Concurrently, for classification, we model the scores of output instances as continuous distributions rather than deterministic values.

Furthermore, we propose three design guidelines for scoring distributions that enable DAP-loss to obtain appropriate gradients for gradient descent, thereby enhancing overall performance. We provide a theoretical proof demonstrating that DAP-loss generates suitable gradients for model training. Extensive experiments on the COCO dataset show that replacing the original loss functions with DAP-loss in the later stages of model training significantly improves performance.

The main contributions of this paper are summarized as follows:

1. We conduct a comprehensive analysis of AP metric, clarifying why it is not suitable for direct gradient descent optimization in object detection models;

2. Building on this analysis, we propose Differentiable Average Precision Loss (DAP-loss), an approximation method for the AP metric. Additionally, we have designed specifically a cost function for the Hungarian matching algorithm to integrate seamlessly with DAP-loss.

3. We provide a theoretical proof demonstrating that DAP-loss ensures mathematical convergence and effectiveness in model training. Furthermore, extensive empirical experiments confirm that DAP-loss enhances the performance of end-to-end object detection models.

## 2 RELATED WORK

### 2.1 DETR FOR OBJECT DETECTION.

The pioneering work DETR(Carion et al., 2020) employed transformers and set-based prediction to achieve end-to-end object detection. Its simplicity and outstanding performance have led to numerous proposed extensions. Deformable DETR(Zhu et al., 2020) introduced a multi-scale deformable self/cross-attention mechanism that selectively focuses on a small number of key sampling points in the reference bounding boxes. Compared to DETR, Deformable DETR significantly accelerates convergence and achieves improved performance. DAB-DETR(Liu et al., 2022) and DN-DETR(Li et al., 2022) demonstrated that the query formulation in the decoder can significantly impact DETR's performance. DINO-DETR(Zhang et al., 2023) achieved shorter training times and better performance, by addressing the instability issues of the one-to-one matching problem. RT-DETR(Zhao et al., 2024), on the other hand, has extended DETR into the realm of real-time object detection, enabling broader practical applications. Building on these, this paper integrates DAP-loss with DETR and its variants, further enhancing their detection capabilities.

### 2.2 AP AS A LOSS FOR OBJECT DETECTION

Average Precision (AP), which takes into account both classification and localization tasks, is the most commonly used evaluation metric in object detection. However, due to its non-differentiability and non-convexity, AP cannot be directly used as an optimization objective in object detection. Several methods have been proposed to tackle the challenge of optimizing AP loss in object detection. AP-loss(Chen et al., 2019) and its extensions(Pu et al., 2024; Xu et al., 2022) utilize error-driven updates, employing Rankloss as a classification loss to indirectly optimize AP. Methods such as Song et al. (2016); Henderson & Ferrari (2017b) use differentiable approximations of AP as training losses. Alternatively, Mohapatra et al. (2018) leverages reinforcement learning to improve the original classification loss starting from AP. Although these methods have achieved some valuable results in optimizing AP loss, they still have limitations. A key issue is that they primarily replace the classification loss with one based on the AP metric, without adequately addressing the localization task. Consequently, these approaches do not directly optimize AP; instead, they rely on regression loss functions and their associated balancing parameters for model training. In contrast, the DAP-loss we propose is a differentiable approximation of AP that can simultaneously optimize both localization and classification tasks without the need for additional balancing hyperparameters.

## 3 METHOD

We aim to directly optimize the AP (Average Precision) metric in end-to-end detectors such as DETR(Carion et al., 2020).

### 3.1 PRELIMINARY

**DETR:** Detection Transformers (DETR) passes the input image $I$ through a backbone network and a transformer encoder to obtain a series of enhanced feature embeddings $X$. These feature embeddings are then combined with a set of $N$ object query embeddings $Q$ and passed through a transformer decoder to produce $N$ output predictions. Finally, DETR performs a one-to-one bipartite matching between the predictions and the ground truth annotations $G$ for bounding boxes and labels, associating each ground truth annotation with the prediction that has the minimum matching cost. Predictions matched to ground truth annotations are classified as **positive** samples, while unmatched ones are considered **negative**. Similarly, our proposed method also classifies samples as positive or negative based on these matching results.

### 3.2 AP METRIC

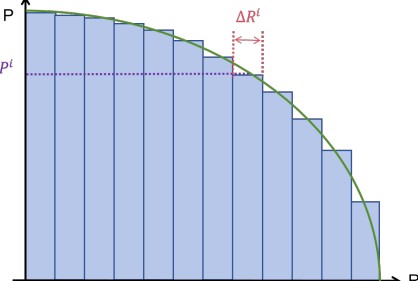

Figure 1: **Rectangle Integration for AP Approximation.** This figure demonstrates the approximation of the area under the Precision-Recall (PR) curve using $N$ rectangles. The precision of the approximation improves as $N$ increases.

The AP in object detection serves as a critical benchmark for evaluating the performance of a detector, derived from the area under the Precision-Recall (PR) curve (averaged over uniformly sampled IoU thresholds ranging from 0.50-0.95 with a step size of 0.05).

As shown in Figure 1, we approximate the definite integral of the PR curve with rectangular method.

$$AP_\alpha = \int_0^1 P_\alpha \, dR_\alpha \approx \sum_{i=0}^{N-1} P_\alpha^i \Delta R_\alpha^i = \frac{1}{G} \sum_{i=0}^{N-1} P_\alpha^i \Delta T_\alpha^i \tag{1}$$

where $\alpha$ represents the IoU threshold, and $\frac{i}{N}$ denotes the score threshold; The value $G$ indicates the total count of ground truth, while $T^i$ and $P^i$ respectively denote the count of true positives and the precision at the score threshold corresponding to $\frac{i}{N}$. $\Delta R^i$ is defined as ($R^{i+1}$ - $R^i$) and $\Delta T^i$ as ($T^{i+1}$ - $T^i$).

With this, AP can be calculated as follows:

$$AP = \frac{1}{10}(AP_{50} + AP_{55} + \ldots + AP_{90} + AP_{95}) \tag{2}$$

$$\approx \frac{1}{10G} \sum_{i=0}^{N-1} (P_{50}^i \Delta T_{50}^i + P_{55}^i \Delta T_{55}^i + \ldots + P_{90}^i \Delta T_{90}^i + P_{95}^i \Delta T_{95}^i) \tag{3}$$

$$= \frac{1}{10G} \sum_{i=0}^{N-1} \sum_{b \in Pos} L(b_{iou}, i) \times H(\frac{i}{N}, b_{score}) \tag{4}$$

where $Pos$ is the set of positive outputs, *i.e.*, those matched with ground truth annotations; and $L(\cdot)$ represents the localization score function of the predicted bounding boxes. The classification function $H(\frac{i}{N}, b_{score})$ is defined as $H(\frac{i}{N}, b_{score}) = 1$ if $\frac{i}{N} \le b_{score} < \frac{i+1}{N}$ and $H(\frac{i}{N}, b_{score}) = 0$ otherwise.

As shown in Equation 4, the AP is determined by the localization score function $L(\cdot)$ and the classification function $H(\cdot)$. In following sections, we will discuss these functions $L(\cdot)$ and $H(\cdot)$ from the perspectives of localization and classification. Additionally, we will introduce the method proposed in this paper for optimizing these functions using backpropagation.

### 3.2.1 LOCALIZATION

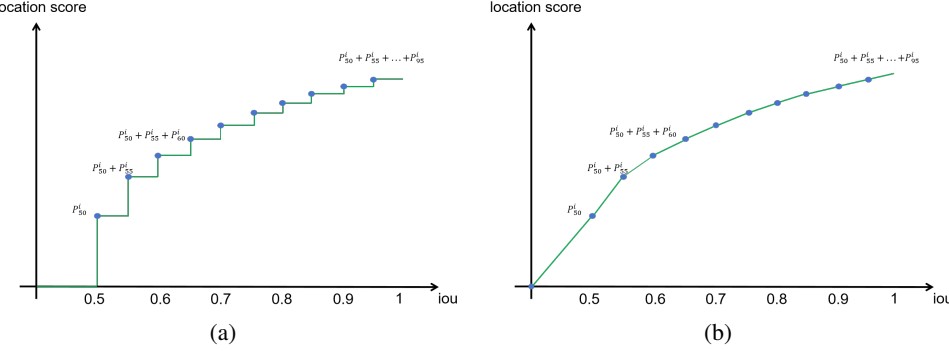

(a)  (b)

Figure 2: **Schematic of the Localization Score Function.** (a) Shows $L(\cdot)$ from Equation 4, where the gradient with respect to IoU is either zero or undefined. (b) Illustrates our proposed $\hat{L}(\cdot)$, where the gradient with respect to IoU is positive.

From Equation 4, it is evident that the localization task pertains to function $L(\cdot)$ and is independent of $H(\cdot)$. Therefore, in this subsection, we focus solely on the function $L(\cdot)$.

Consider a predicted bounding box matched with a ground truth (GT), which serves as a positive sample during training. Under a score threshold of $\frac{i}{N}$, the visualization of function $L(\cdot)$ is shown in Figure 2(a).

$L(\cdot)$ is determined by 10 points: $(0.5, P_{50}^i), (0.55, P_{50}^i + P_{55}^i), \ldots, (0.95, P_{50}^i + P_{55}^i + \ldots + P_{95}^i)$. It is clear that within the range of $IoU \in [0, 1]$, the gradient of function $L(\cdot)$ with respect to IoU is zero. Consequently, the task of optimizing localization cannot be effectively addressed using the backpropagation algorithm.

As shown in Figure 2(b), we propose using an interpolation technique to address the aforementioned gradient deficiency issue. Specifically, we introduce an additional point at the origin (0, 0) and extend the line segment from 0.9 to 0.95 up to 1.0, thereby ensuring that the function yields reasonable values for $IoU$ within the ranges [0,0.5] and [0.95,1]. We denote this interpolated function as $\hat{L}(\cdot)$. As illustrated in the figure, $\hat{L}(\cdot)$ provides a reasonable gradient across the entire IoU range [0, 1].

### 3.2.2 Classification

For classification task, it is intuitive that the precision in $L(\cdot)$ and the step function $H(\cdot)$ cannot produce trainable gradients. This limitation arises because the classification scores are discrete, whereas gradient descent algorithms require smooth functions.

To derive an appropriate gradient, we employ a continuous probabilistic distribution to model the scores of the predicted bounding boxes. Given a score threshold $x$, the probability that a predicted bounding box $b$ is classified as positive is represented by the **complementary cumulative distribution function** (tail distribution), defined as follows:

$$Pr(b, x) = \int_x^1 f^b(y)dy \tag{5}$$

where $f^b(\cdot)$ is the probabilistic distribution of $b$. When $f^b(\cdot)$ is represented as an impulse function, the equation above becomes equivalent to the original model. Therefore, the significance of $H(\cdot)$ in Equation 4 can be interpreted as the cumulative distribution function of the impulse function $\delta(\cdot)$ over the interval from $\frac{i}{N}$ to $\frac{i+1}{N}$.

To establish design guidelines for $f(\cdot)$, we calculated the partial derivatives of the AP with respect to the scores of positive and negative samples, yielding the following equations. The detailed derivation can be found in Appendix A.3:

$$\frac{\partial AP_\alpha}{\partial s_t} = \frac{1}{G} \int_0^1 \frac{\partial f^t(x)}{\partial s_t} \times P(x) + \frac{\frac{\partial Pr(t,x)}{\partial s_t} \times fp(x)}{(tp(x) + fp(x))^2} \times \sum_{m \in tp} f^m(x)dx \tag{6}$$

$$\frac{\partial AP_\alpha}{\partial s_n} = \frac{1}{G} \int_0^1 \frac{-\frac{\partial Pr(n,x)}{\partial s_n} \times tp(x)}{(tp(x) + fp(x))^2} \times \sum_{m \in tp} f^m(x)dx \tag{7}$$

where $fp(x)$ and $tp(x)$ respectively represent the false positive and true positive rates at a score threshold of $x$ and an IoU threshold of $\alpha$. Additionally, $s_t$ and $s_n$ denote the scores of a positive sample and a negative sample, respectively.

In the original model (*i.e.*, $f(x)$ is $\delta(\cdot)$), both $\frac{\partial Pr(t,x)}{\partial s_t}$ and $\frac{\partial Pr(n,x)}{\partial s_n}$ are zero except at a finite number of points. This characteristic significantly affects the gradients $\frac{\partial AP_\alpha}{\partial s_t}$ and $\frac{\partial AP_\alpha}{\partial s_n}$, causing the derivatives of the classification scores to become zero.

To facilitate training with the gradient descent algorithm, we aim for the function $f(\cdot)$ to satisfy the following characteristics: *i.* $\frac{\partial AP_\alpha}{\partial s_t} > 0$, $\frac{\partial AP_\alpha}{\partial s_n} < 0$; *ii.* $f(\cdot)$ is an approximation of $\delta(\cdot)$.

For the first characteristic, a simple and effective approach is to ensure that each term in Equation 6 is greater than or equal to 0, while each term in Equation 7 is less than or equal to 0. We obtain the following specific conditions that enable $f(\cdot)$ satisfy this characteristic. The relevant mathematical proof can be found in Appendix A.2 of this paper.

The conditions for the distribution $f(\cdot)$ are as follows:

1. $f(x) \geq 0$ for all $x \in [0, 1]$;
   *Condition 1* ensures that the distribution is non-negative across the entire range.

2. $\int_0^1 f(x)\,dx = 1$;
   *Condition 2* ensures that the distribution is properly normalized, such that the total probability sums to 1.

3. $\int_a^1 f^{b_1}(x)\,dx > \int_a^1 f^{b_2}(x)\,dx$ for $s_1 > s_2$ and for all $a \in (0, 1)$.
   *Condition 3* ensures that a bounding box with a higher score $s_1$ will have a greater probability of being a true positive compared to a box with a lower score $s_2$, thereby preserving the ranking of boxes based on their scores.

For the second characteristic, which states that $f(\cdot)$ is an approximation of $\delta(\cdot)$, we use a normalized Gaussian function to approximate $\delta(\cdot)$, setting the mean to the classification score $s$ and using a constant standard deviation $\sigma$. Thus, in our proposed method, we use $F(\frac{i}{N}, b_{score})$ to replace $H(\cdot)$

from Equation 4. The characteristics of $F(\cdot)$ and $H(\cdot)$ are discussed in more detail in Appendix A.1. The definition of $F(\frac{i}{N}, b_{score})$ is as follows:

$$F(\frac{i}{N}, b_{score}) = \begin{cases} \int_{-\infty}^{\frac{1}{N}} \mathcal{N}(x; b_{score}, \sigma) dx & \text{if } i = 0, \\ \int_{\frac{i}{N}}^{\frac{i+1}{N}} \mathcal{N}(x; b_{score}, \sigma) dx & \text{if } 0 < i < N - 1, \\ \int_{\frac{i}{N}}^{+\infty} \mathcal{N}(x; b_{score}, \sigma) dx & \text{if } i = N - 1. \end{cases} \quad (8)$$

In summary, the differentiable approximation of Equation 4, referred to as DAP-loss in this paper, can be expressed as follows:

$$\text{DAP-loss} = -1 \times DAP \quad (9)$$

$$= \frac{-1}{10G} \sum_{i=0}^{N-1} \sum_{b \in Pos} \hat{L}(b_{iou}, i) \times F(\frac{i}{N}, b_{score}) \quad (10)$$

where $\hat{L}(\cdot)$ represents a differentiable localization score function, as elaborated in Subsection 3.2.1. $F(\frac{i}{N}, b_{score})$ denotes the cumulative distribution function from $\frac{i}{N}$ to $\frac{i+1}{N}$, based on the probability density function of $f^b(\cdot)$.

## 3.3 AP-Cost Matcher

In general, DETR and its variants use the same function to define the Hungarian matching cost function $L_{matcher}$ as that used for the model training loss. Consistent with previous approaches, we will next introduce a Hungarian matching cost function designed to accommodate the DAP-loss.

In end-to-end object detection models, the positive and negative samples during training are determined by the output of the matcher. This means that when computing the matching cost $L_{matcher}$, the information regarding which samples are positive is unavailable, making it impossible to compute $P_i^{50}, P_i^{55}, \ldots, P_i^{95}$ in Equation 3.

In this paper, we introduce two approaches to address the issues previously discussed, *i.* We have devised a method that calculates the cost based on the outcomes of preceding iterations and performs momentum updates. Specifically, we employ the following formula to update the momentum parameter $\mathcal{P}_\alpha^i = m\mathcal{P}_\alpha^i + (1 - m)P_\alpha^i$, where $\alpha \in \{0.5, 0.55, \ldots, 0.95\}$; *ii.* The second approach simplifies the process by assigning the constant $\frac{1}{N}$ to all instances of $\mathcal{P}_i^\alpha$.

For a predicted bounding box $b$ and a ground truth $g$, the cost function for the bipartite matching can be expressed as,

$$L_{matcher}(b, g) = \sum_{i=0}^{N-1} \hat{L}(IoU(b, g), i) \times F(\frac{i}{N}, b_{score}) \quad (11)$$

It is important to note that computing $P_\alpha^i$ within $\hat{L}$ is challenging, so it is replaced by $\mathcal{P}_\alpha^i$ as discussed above.

## 3.4 Detail of Training Algorithm

**Minibatch Training:** The minibatch training strategy is commonly employed in deep learning frameworks (Krizhevsky et al., 2012; Vaswani, 2017; Chen et al., 2019), offering better stability and faster convergence compared to using a batch size of 1. Minibatch training is pivotal for our optimization algorithm due to the batch size's notable influence on AP calculation. An excessively small batch can cause a significant discrepancy between the estimated and actual AP values. For example, consider an extreme case where our model predicts perfect rankings and localization results for images $I1$ and $I2$, but the lowest score in $I1$ is even higher than the highest score in $I2$. In such a case, both $I1$ and $I2$ would individually yield very high AP values. Aggregating image scores within a minibatch helps to avoid this issue. In the next section, we will present experimental results demonstrating the impact of batch size on our method.

---

**Algorithm 1** Computation of DAP-loss

---

**Input:** A batch of training data $I$, label $y$, and model $F_\theta$
**Output:** DAP-loss of input data
1: $\hat{N} \leftarrow \{0, 1, 2, \ldots, N-1\}$;
2: $Pred \leftarrow F_\theta(I)$;                           ▷ Predict outputs using the model
3: Compute $F(\frac{i}{N}, b_{score})$, for all $b \in Pred$, $i \in \hat{N}$;        ▷ According to Equation 8
4: $Pos, Neg \leftarrow$ Matcher$(Pred, y)$;                 ▷ Cost function as per 11
5: $G \leftarrow Pos.len()$;                       ▷ Number of positive samples
6: Compute $P_\alpha^i$, for all $\alpha \in \{0.5, 0.55, \ldots, 0.95\}$, $i \in \hat{N}$;     ▷ Precision values
7: Compute $\hat{L}(b_{iou}, i)$, for all $b \in Pos$, $i \in \hat{N}$;     ▷ Function $\hat{L}$ as per Subsection 3.2.1
8: DAP-loss $\leftarrow \frac{-1}{10G} \sum_{i \in \hat{N}} \sum_{b \in Pos} \hat{L}(b_{iou}, i) \times F(\frac{i}{N}, b_{score})$    ▷ According to Equation 10
9: **Return:** DAP-loss

---

**Interpolated AP:** Interpolated Average Precision (Interpolated AP) is widely used in object detection benchmarks, such as PASCAL VOC (Everingham et al., 2015) and MS COCO (Lin et al., 2014). Compared to standard AP, Interpolated AP is less sensitive to minor fluctuations in predicted scores and better aligns with practical needs, which is why it is more commonly employed today. For these reasons, we adopt Interpolated AP instead of the original version. This implies that precision increases as the score threshold increases, *i.e.*, $P(i) \leq P(j)$ if $i < j$.

The algorithmic details and computation process of DAP-loss are summarized in Algorithm 1. Our method does not require any additional loss functions for training. Therefore, using backpropagation with DAP-loss to update the model completes a training iteration.

## 4 EXPERIMENT

**Dataset:** In this paper, we systematically conduct all experiments by training our models on the COCO 2017 (Lin et al., 2014) training dataset (118K images), which is a widely recognized benchmark in the field. We rigorously evaluate the performance of these models using the COCO 2017 validation set (5K images), to ensure that our results are both reliable and comparable to existing literature. Unless otherwise specified, we report AP as COCO-style (Lin et al., 2014) bbox AP, which is the integral metric over multiple thresholds.

**Experiments Setting:** We evaluate the proposed method against several state-of-the-art (SOTA) approaches (Carion et al., 2020; Zhao et al., 2024; Zhu et al., 2020) using ResNet-50 and ResNet-101 (He et al., 2016) backbones. Unless otherwise specified, all experimental parameters follow the settings outlined below.

The learning rate for all experiments was set to one-tenth of that used in the baseline method and further reduced to one-hundredth after 7 epochs. Due to GPU memory constraints, batch sizes were configured as follows: 40 for DETR and RT-DETR with the ResNet-101 backbone, 56 for the ResNet-50 backbone, and 24 for the deformable model also using the ResNet-50 backbone.

Aside from the learning rate and batch size, all other parameters are consistent with those of the original methods. The standard deviation ($\sigma$) of the distribution $\mathcal{N}$ in the DAP-loss is uniformly set to 0.05 across all experiments, and the variable **G** in Equation 4 is consistently assigned a value of 256. All experimental results were obtained from models that underwent 12 epochs of fine-tuning according to these settings following their initial training.

### 4.1 EXPERIMENTS BASED ON DIFFERENT MODLES

Table 1 provides a detailed comparison between our proposed method and several widely recognized models.

**Experiments based on DETR:** Experimental evaluations show that, using ResNet-50 and ResNet-101 backbones, our proposed DAP-loss improves the Average Precision (AP) by 1.2% and 1.3%, respectively, compared to the original method. The AP gains are notably higher at stricter Intersection over Union (IoU) thresholds, surpassing the baseline loss by 0.5% and 0.2% in $AP_{50}$ with

Table 1: Detection results on COCO validation set 2017. The term 'Raw Loss' refers to the original loss function used by the method, with all parameters remaining consistent. '†' denotes that IoU reward-based query selection (Zhao et al., 2024) was not used.

| Method | Backbone | DAP Loss | Raw Loss | AP | $AP_{50}$ | $AP_{75}$ |
|---|---|---|---|---|---|---|
| DETR | R50 | | ✓ | 42.2 | 62.5 | 44.6 |
| DETR | R50 | ✓ | ✓ | 42.7 | 63.0 | 45.1 |
| DETR | R50 | ✓ | | 43.4(+1.2) | 63.0 | 45.8 |
| DETR | R101 | | ✓ | 43.6 | 64.1 | 46.1 |
| DETR | R101 | ✓ | | 44.9(+1.3) | 64.3 | 48.0 |
| deformable DETR | R50 | | ✓ | 46.2 | 64.9 | 50.2 |
| deformable DETR | R50 | ✓ | | 47.3(+1.1) | 65.1 | 51.3 |
| RT-DETR† | R50 | | ✓ | 52.2 | 71.1 | 56.8 |
| RT-DETR† | R50 | ✓ | | 52.8(+0.6) | 71.2 | 57.3 |
| RT-DETR | R50 | | ✓ | 53.1 | 71.4 | 57.5 |
| RT-DETR | R50 | ✓ | ✓ | 53.3(+0.2) | 71.7 | 57.5 |

ResNet-50 and ResNet-101, respectively, and by 1.2% and 1.9% in $AP_{75}$. When using DAP-loss in conjunction with raw-loss in the DETR(Carion et al., 2020) framework with a ResNet-50 backbone, we achieve an AP of 42.7%, representing a 0.5% increase over raw-loss alone and a 0.7% decrease compared to DAP-loss used alone. This finding highlights the effective balance DAP-loss strikes between classification and localization, resulting in improved overall performance.

**Experiments based on deformable DETR:** Table 1 presents the experimental outcomes for deformable DETR(Zhu et al., 2020) employing our proposed DAP-loss with a ResNet-50 backbone. The results show that DAP-loss outperforms raw-loss by 1.1% in AP, 0.2% in $AP_{50}$, and 1.1% in $AP_{75}$.

**Experiments based on RT-DETR:** In Table 1, entries marked with a superscript '†' denote the absence of IoU-aware query selection techniques (Zhao et al., 2024). The experimental results show that without query selection, DAP-loss achieves a 0.6% improvement in AP over the baseline, demonstrating its efficacy. However, with query selection, DAP-loss provides only a slight advantage, highlighting the effectiveness of the query selection. This design is tightly integrated with the raw loss and cannot be separated. Consequently, we are unable to conduct experiments involving query selection without raw loss.

## 4.2 ABLATION STUDY

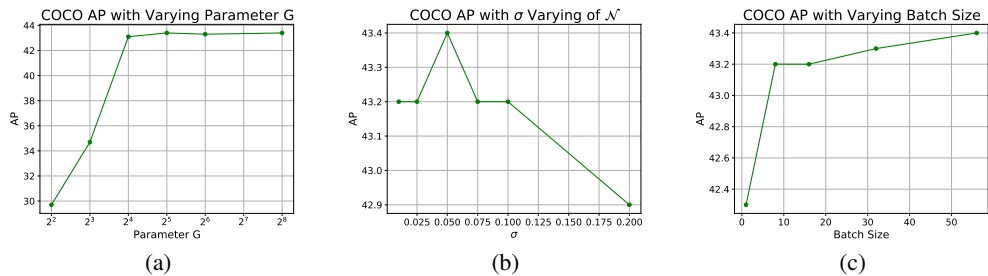

(a)    (b)    (c)

Figure 3: **Ablation experiments.** COCO validation 2017 results based on DETR.

In this subsection, we conduct a systematic analysis to assess the impact of different components and experimental settings of our method. Unless otherwise specified, all experimental configurations in this subsection adhere to those used in the DETR experiments described earlier, with the backbone network fixed as ResNet-50.

**Ablations on parameter $N$:** In Equations 1 and 4, the parameter $N$ denotes the number of rectangles used to segment the area beneath the Precision-Recall (PR) curve. It is evident that as $N$ increases, the method proposed in this paper provides a closer approximation to AP.

Figure 3(a) illustrates how the COCO AP varies with $N$. The trend indicates a general increase in AP as $N$ rises, with stabilization occurring beyond a value of 32. This behavior aligns with our earlier discussion, indicating that a higher Parameter $N$ leads to a more accurate approximation of the true AP for a set of images.

**Ablations on standard deviation $\sigma$ of $\mathcal{N}$:** We examined the standard deviation $\sigma$ of $\mathcal{N}$, the sole hyperparameter in DAP-loss that requires fine-tuning. A very large $\sigma$ can cause the function to deviate from the properties of the step function $H(\cdot)$ in Equation 4, while a very small $\sigma$ can result in excessively small gradients, potentially hindering training. Figure 3(b) presents COCO AP results for various $\sigma$ values, indicating that the best performance is achieved at $\sigma = 0.05$, with satisfactory results within the range of [0.01, 0.1]. The differences between function $H(\cdot)$ and function $F(\cdot)$ under various values of $\sigma$ can be found in Appendix A.1.

**Ablations on batch size:** As depicted in Figure 3(c), optimal performance generally increases with batch size within the range of [1, 56]. Notably, at a batch size of 8, the performance is only 0.2% AP lower than the best result. This trend is intuitive, as a very small batch size can lead to a significant discrepancy between the AP of the batch and that of the entire dataset.

Table 2: Results of different matching functions

| Backcone | Cost Function Mode | AP |
|---|---|---|
| R50 | raw | 42.8 |
| R50 | momentum | 43.4 |
| R50 | constant | 43.4 |
| R101 | raw | 44.6 |
| R101 | constant | 44.9 |

Table 3: Results of training models from scratch.

| raw loss | DAP-loss | AP |
|---|---|---|
| ✓ | | 39.5 |
| | ✓ | 29.4(-6.1) |
| ✓ | ✓ | 35.8(- 3.7) |

**Ablations on the Hungarian Matcher cost function:** In Subsection 3.3, we introduced two matching cost functions designed for DAP-loss. Table 2 reports the results obtained with these different matching functions. Both the Constant and Momentum modes show similar improvements and outperform the Raw mode. Considering the greater complexity of the Momentum model, we recommend using the Constant mode.

**Results of training models from scratch:** As shown in Table 3, the proposed DAP-loss results in a decrease in model performance, whether used alone or in combination with the original loss. We propose two possible reasons for this decline: 1) DAP-loss is computed at the batch-level, which may increase the likelihood of overfitting compared to the instance-level raw loss; and 2) when prediction results are poor, the gradients of DAP are too small. Although DAP-loss has the potential to enhance final performance, it is less effective for training from scratch.

## 5 CONCLUSION

In this paper, we decompose Average Precision (AP) into the product of a localization function and a classification function, analyzing their non-differentiability and non-convexity. We then employ interpolation and Gaussian-like smoothing techniques to develop a differentiable approximation of AP, termed Differentiable Average Precision (DAP). Furthermore, we extend DAP-loss to the matching cost of the Hungarian algorithm, making it suitable for end-to-end detection models. DAP-loss optimizes both localization and classification tasks simultaneously, and effectively balancing these two objectives without the need for hyperparameter tuning. We provide a solid theoretical analysis of the proposed DAP-loss, and experimental results demonstrate that DAP-loss enhances the final performance of trained models, although it is not suitable for training from scratch.

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

# A APPENDIX

## A.1 VISUAL COMPARISON OF FUNCTION $F$ AND $H$

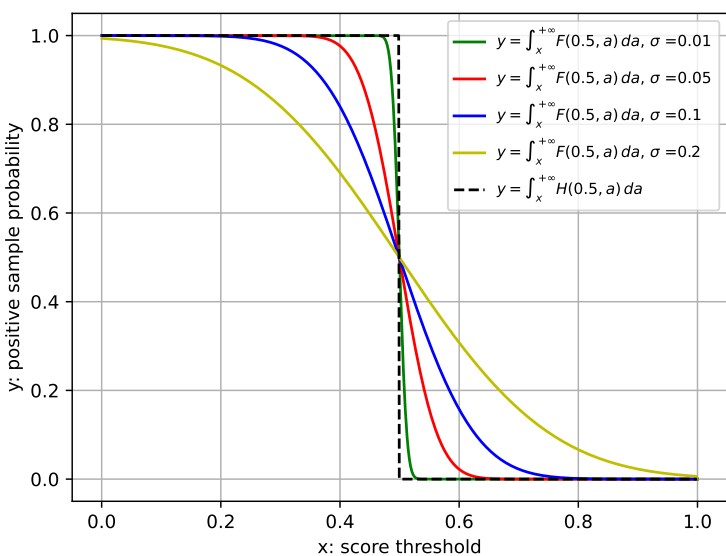

Figure 4: Under a prediction box score of 0.5, the probability of positive samples (y-axis) corresponding to different score thresholds (x-axis). The dashed line represents the function $H(\cdot)$ from Equation 4, while the solid line represents the function $F(\cdot)$ from Equation 10.

Figure 4 illustrates the characteristics of the function $H(\cdot)$ and the function $F(\cdot)$ under various sigma values. As sigma decreases, $F(\cdot)$ becomes more similar to $H(\cdot)$, though this also exacerbates the issue of gradient vanishing. Experiments indicate that sigma values in the range of [0.01, 0.1] yield favorable results.

## A.2 PARTIAL DERIVATIVE OF DAP WITH RESPECT TO CLASSIFICATION SCORES

We provide a proof for the proposition mentioned in Section 3.2.2 of this paper. It is important to note that the DAP presented in this section is equivalent to the AP in Equations 6 and 7. Additionally, the equations in this section consider only cases where $tp > 0$. As shown in Equation 7:

$$G \times \frac{\partial AP}{\partial s_n} = \int_0^1 \frac{-\frac{\partial Pr(n,x)}{\partial s_n} \times tp(x)}{(tp(x) + fp(x))^2} \times \sum_{m \in tp} f^m(x) dx \tag{12}$$

where, $G$ is the number of gt labels. According to Condition 2, $f(x)$ is greater than 0, and both $tp(x)$ and $fp(x)$ are greater than or equal to 0. According to Condition 3, $\frac{\partial Pr(n,x)}{\partial s_n} > 0$. Hence,

$$\frac{\partial AP}{\partial s_n} < 0 \tag{13}$$

As shown in Equation 6:

$$G \times \frac{\partial AP}{\partial s_t} = \int_0^1 \frac{\partial f^t(x)}{\partial s_t} \times P(x) + \frac{\frac{\partial Pr(t,x)}{\partial s_t} \times fp(x)}{(tp(x) + fp(x))^2} \times \sum_{m \in tp} f^m(x) dx \tag{14}$$

According to Condition 2 and Condition 3 in Section 3.2.2, $f^m(x) > 0$, so $\frac{\frac{\partial Pr(t,x)}{\partial s_t} \times fp(x)}{(tp(x)+fp(x))^2} \times \sum_{m \in tp} f^m(x) > 0$. Therefore,

$$G \times \frac{\partial AP}{\partial s_t} > \int_0^1 \frac{\partial f^t(x)}{\partial s_t} \times P(x) dx \tag{15}$$

applying integration by parts to the above equation yields:

$$G \times \frac{\partial AP}{\partial s_t} > (P(x) \times \int_0^x \frac{\partial f^t(y)}{\partial s_t} dy)]_0^1 - \int_0^1 \frac{\partial P(x)}{\partial x} \int_0^x \frac{\partial f^t(y)}{\partial s_t} dy dx \tag{16}$$

$$= -\int_0^1 \frac{\partial P(x)}{\partial x} \int_0^x \frac{\partial f^t(y)}{\partial s_t} dy dx \tag{17}$$

$$= -\int_0^1 \frac{\partial P(x)}{\partial x} \frac{\int_0^x f^t(y) dy}{\partial s_t} dx \tag{18}$$

since we use Interpolated AP, we have $\frac{\partial P(x)}{\partial x} \geq 0$. According to Condition 3 in Section 3.2.2, we obtain $\frac{\int_0^x f^t(y) dy}{\partial s_t} < 0$. In summary, we have proved that:

$$\frac{\partial AP}{\partial s_t} > 0 \tag{19}$$

Based on Equations 13 and 19, it is evident that our proposed DAP is feasible for optimizing classification tasks.

## A.3 PARTIAL DERIVATIVE OF AP

To analyze the properties of AP, we compute the partial derivatives of AP with respect to the classification scores. Here, $s_t$ represents the score of a positive sample and $s_n$ represents the score of a negative sample. Based on Equation 1, we compute the gradient of the $AP$ with respect to the classification scores.

$$\frac{\partial AP}{\partial s_t} = \frac{\int_0^1 P(x) \frac{\partial R(x)}{\partial x} dx}{\partial s_t} \tag{20}$$

$$= \frac{\int_0^1 P(x) \times \sum_{m \in tp} f^m(x) dx}{G \times \partial s_t} \tag{21}$$

$$= \frac{1}{G} \times \int_0^1 \frac{\partial P(x)}{\partial s_t} \times \sum_{m \in tp} f^m(x) + P(x) \frac{\partial f^t(x)}{\partial s_t} dx \tag{22}$$

$$= \frac{1}{G} \int_0^1 \frac{\partial f^t(x)}{\partial s_t} P(x) + \frac{\frac{\partial Pr(t,x)}{\partial s_t} \times fp(x)}{(tp(x) + fp(x))^2} \times \sum_{m \in tp} f^m(x) dx \tag{23}$$

$$\tag{24}$$

$$\frac{\partial AP}{\partial s_n} = \frac{\int_0^1 P(x)\frac{\partial R(x)}{\partial x}dx}{\partial s_n} \tag{25}$$

$$= \frac{\int_0^1 P(x) \times \sum_{m \in tp} f^m(x)dx}{G \times \partial s_n} \tag{26}$$

$$= \frac{1}{G} \times \int_0^1 \frac{\partial P(x)}{\partial s_n} \times \sum_{m \in tp} f^m(x) + P(x)\frac{\partial f^t(x)}{\partial s_n}dx \tag{27}$$

$$= \frac{1}{G} \int_0^1 \frac{-\frac{\partial Pr(n,x)}{\partial s_n} \times tp(x)}{(tp(x) + fp(x))^2} \times \sum_{m \in tp} f^m(x)dx \tag{28}$$

where $Pr(t, x) = \int_x^1 f^t(y)dy$ represents the probability that the predicted bounding box $t$ is a positive sample given a score threshold $x$.

