# OpenReview forum: "Differentiable Average Precision Loss in DETR"
_ICLR.cc/2025/Conference — ICLR 2025 Conference Withdrawn Submission_

### Official Review · Reviewer_be3V · 2024-10-27

**Soundness:** 2
**Presentation:** 2
**Contribution:** 1
**Rating:** 1
**Confidence:** 5

**Summary:**

In this paper, the authors study the problem of training an object detector by using the performance measure (Average Precision -- AP) as the training objective. This is not the first time that this is attempted. Compared to what has been studied in the literature before, the authors consider mAP, the average of AP over different IoU thresholds. This sets the paper significantly apart from prior work since it includes both classification and localization aspects. mAP is not differentiable, similar to AP. Therefore, the authors introduce some approximations to obtain the proposed loss's derivatives.

The loss function is used to train DETR-based detectors and evaluated on the COCO dataset. Some noticable improvements are reported over DETR, despite unpromising results on other detectors.

**Strengths:**

+ It is worthwhile to use performance measures as training objectives for object detectors.
+ It is promising to adapt mAP as a loss function.

**Weaknesses:**

Although I am extremely fond of ranking-based training of object detectors, I strongly believe that the paper is far from being ready for a top-level conference. The main reasons include:

1. First of all, the paper's formulations and derivations include major gaps / flaws and require substantial fixing:

1.1. Eq 3 to 4: The transition to Eq 4 is not clear and appears to be flawed by my calculation. To see this, note the following: P values (in Eq 3) are between 0-1 whereas "Delta T" are counts of TPs. On the other hand, Eq 4 multiplies localization score (between 0 and 1) with H() values (between 0 and 1). These multiplications in Eq 3 and 4 yield different ranges of values.

1.2. Eq 4: "From Equation 4, it is evident that the localization task pertains to function L(·) and is independent of H(·)." => Not sure how this can be claimed given that the equation does depend/include H().

1.3. The derivation of the gradient in Section 3.2.2 is flawed since the derivation (in Appendix 3) (i) ignores the dependence of P(x) on x and s_t, (ii) somehow skips very critical intermediate steps from Eq 3 to Eq 20, and (iii) includes an unexpected term dR(x)/dx.

1.4. Line 304: There is a flaw in the momentum update: P^i_a = m P^i_a + (1-m) P^i_a.

1.5. Line 311: "computing Piα within ˆL is challenging, so it is replaced by Piα" => There is a flaw here.

2. The paper misses major prior work that has significant overlap with the paper's motivations and contributions. Some include:

2.1. The paper misses a very fundamental loss formulation that has the same goals and overall motivation as the proposed DAP loss:

Oksuz, K., Cam, B. C., Akbas, E., & Kalkan, S. (2020). A ranking-based, balanced loss function unifying classification and localisation in object detection. Advances in Neural Information Processing Systems, 33, 15534-15545.

2.2. For AP Loss with DETR, look at the following paper which has applied a ranking-based loss (combining classification & localization) into DETR-based detectors:

F. Yavuz, B. C. Cam, A. H. Dogan, K. Oksuz, E. Akbas*, S. Kalkan*, "Bucketed Ranking-based Losses for Efficient Training of Object Detectors", 18th European Conference on Computer Vision (ECCV), 2024.

2.3. The paper also misses the works (several papers) from the group of Georg Martius.

3. The paper's contributions are questionable.

3.1. Contribution 1: Comprehensive analysis of the AP metric => The analysis is hardly comprehensive.

3.2. Contribution 2: Differentiable AP Loss => Considering the missing references and positioning with respect to prior work, this is questionable.

3.3. Contribution 3: Proof of theoretical convergence => As highlighted above, there very major flaws in the derivations, which makes this questionable.

4. Major issues in the methodology:

4.1. "Extensive experiments on the COCO dataset show that replacing the original loss functions with DAP-loss in the later stages of model training significantly improves performance." => This is a very significant weakness of the proposed approach, considering that similar approaches (AP Loss, aLRP Loss, RS Loss) can train object detectors from scratch with significant gains over their baselines.

4.2. Not clear why the cleaner formulation of AP Loss by Chen et al. is not adopted for different IoU thresholds.

4.3. "All experimental results were obtained from models that underwent 12 epochs of fine-tuning according to these settings following their initial training." => It is not clear what this initial training phase includes.

5. Experimental evaluation is not convincing.

5.1. The results in Table 1 suggest that the method fails to introduce any gains over more recent DETR detectors. Moreover, given that the proposed method does not start from scratch, it is not clear whether it is fair to compare the results to a training strategy from scratch.

5.2. No results are provided over Co-DETR.

5.3. No results are provided on non-DETR object detectors.

5.4. No comparisons are provided with other ranking-based loss functions.


Minor comments:

- "AP-loss(Chen et al., 2019)" => Please leave space before an opening paranthesis. This issue appears to repeat across the paper.

- Line 157: "The value G indicates" => On the same page, you used G to denote the ground truths. Please use a different symbol.

- Eq 2: You should specify why you choose these thresholds.

- Eq 2: I would recommend calling this mAP to avoid confusion.

- Eq 4: b_score is not introduced.

- Figure 2: Text too small. "location score" => "Localization score".

- Page 4: L() is not properly defined. As such, it is very difficult to follow the derivations and the arguments.

- Line 228: "where fb(·) is the probabilistic distribution of b." => Not clear what this represents considering that a box (b) can have both location (distribution) and class (distribution).

- Line 229-230: "Therefore, the significance of H(·) in Equation 4 can be interpreted as the cumulative distribution function of the impulse function" => This does not make sense at all.

- Line 242: tp(x) => Previously you used T to denote the number of TPs. Please use the symbols consistently.

- Line 232: f() => f^b()?

- Line 371: "Modles" => "Models".

**Questions:**

Please see Weaknesses.

---

### Official Review · Reviewer_oQBy · 2024-10-30

**Soundness:** 3
**Presentation:** 3
**Contribution:** 3
**Rating:** 5
**Confidence:** 4

**Summary:**

This paper introduces Differentiable Average Precision Loss (DAP-loss), a new loss function that approximates AP in a differentiable way, enabling direct AP optimization. DAP-loss improves on traditional methods by supporting joint optimization of classification and localization in end-to-end models without needing task-balancing hyperparameters. Applied to state-of-the-art models like DETR and Deformable DETR, DAP-loss achieves notable performance gains on the COCO dataset, showing its theoretical and practical effectiveness.

**Strengths:**

- The paper is well-written.
- The joint optimization of classification and localization in end-to-end models is interesting, especially as it eliminates the need for task-balancing hyperparameters.

**Weaknesses:**

1. The main weakness of this paper is the insufficient experimentation:
- There is a lack of experiments using large backbone models (e.g., Swin-L).
- Comparative analysis with other methods designed to optimize AP loss for object detection [1] [2] [3] [4] is missing, which would more accurately demonstrate the effectiveness of DAP-loss.

2. There are also some missing references, such as [1].

3. Moreover, the reduced effectiveness of DAP-loss when training from scratch limits its technical contribution to practical applications.

[1] Chenxin T, Li Z, Zhu X, et al. Searching parameterized AP loss for object detection[J]. Advances in Neural Information Processing Systems, 2021, 34: 22021-22033.

[2] Chen K, Li J, Lin W, et al. Towards accurate one-stage object detection with ap-loss[C]//Proceedings of the IEEE/CVF Conference on Computer Vision and Pattern Recognition. 2019: 5119-5127.

[3] Xu D, Deng J, Li W. Revisiting ap loss for dense object detection: Adaptive ranking pair selection[C]//Proceedings of the IEEE/CVF Conference on Computer Vision and Pattern Recognition. 2022: 14187-14196.

[4] Oksuz K, Cam B C, Akbas E, et al. A ranking-based, balanced loss function unifying classification and localisation in object detection[J]. Advances in Neural Information Processing Systems, 2020, 33: 15534-15545.

**Questions:**

Please supplement the aforementioned experiments. The reviewer is willing to adjust the score accordingly.

---

### Official Review · Reviewer_T24g · 2024-11-01

**Soundness:** 2
**Presentation:** 3
**Contribution:** 2
**Rating:** 3
**Confidence:** 4

**Summary:**

This paper analysis the properties of the AP metric and proposes DAP-loss to provide a differentiable approximation for AP to can optimize it. Additionally, a cost function for the Hungarian matching algorithm is designed to integrate it with DAP-loss. To validate the effect of DAP-loss experiments are conducted on DERT models.

**Strengths:**

The paper is straight forward, easy to understand and well presented. A set of experiments are conducted on COCO dataset with different backbones.

**Weaknesses:**

1-	In am not agree with the “non-differentiable and non-convex nature of AP poses significant challenges” statement. It is correct since AP is non-differentiable and has difficulty to optimise with a gradient-based method. However, AP is a very strong metric and is widely used for evaluating various CV applications.
2-	It is mentioned, “we have designed specifically a cost function for the Hungarian matching algorithm”.  The Hungarian algorithm provides one-to-one matching of predicted bounding boxes to ground-truth annotations during training which is a strict matching and alignment process. How does the proposed cost function address this challenge?
3-	The proposed model is only evaluated on the COCO dataset, which limits to evaluation of the generalization of the proposed model. Some other datasets such as PASCAL-VOC can be considered to evaluate the performance of the model.
4-	Only DETR and RT-DETR are used for evaluation. There are several recent and strong models such as DN-DETR and Rank-DRT that were not used for the evaluation.

**Questions:**

Please see the weaknesses.

---

### Note · Authors · 2024-11-14

I have read and agree with the venue's withdrawal policy on behalf of myself and my co-authors.